# Impact of Thin Meconium on Delivery and Early Neonatal Outcomes

**DOI:** 10.3390/children10020215

**Published:** 2023-01-26

**Authors:** Hanoch Schreiber, Adi Shilony, Reut Batia Amrami, Gal Cohen, Ofer Markovitch, Tal Biron-Shental, Sofia Bauer-Rusek, Shmuel Arnon, Michal Kovo

**Affiliations:** 1Department of Obstetrics and Gynecology, Meir Medical Center, Kfar Saba 4428163, Israel; 2Sackler School of Medicine, Tel Aviv University, Tel Aviv 6329302, Israel; 3Department of Neonatology, Meir Medical Center, Kfar Saba 4428163, Israel

**Keywords:** meconium, thin meconium, meconium aspiration syndrome, adverse neonatal outcomes

## Abstract

Several reports regarding the effects of thin meconium on maternal and neonatal outcomes are contradictory. This study evaluated the risk factors and obstetrical outcomes during deliveries complicated with thin meconium. This retrospective cohort study included all women with a singleton pregnancy, who underwent trial of labor >24 weeks of gestation, in a single tertiary center, over a six-year period. Obstetrical, delivery, and neonatal outcomes were compared between deliveries with thin meconium (thin meconium group) to deliveries with clear amniotic fluid (control group). Included in the study were 31,536 deliveries. Among them 1946 (6.2%) were in the thin meconium group and 29,590 (93.8%) were controls. Meconium aspiration syndrome was diagnosed in eight neonates in the thin meconium group and in none of the controls (0.41%, *p* < 0.001). In multivariate logistic regression analysis, the following adverse outcomes were found to be independently associated with increased odds ratio (OR) for thin meconium: intrapartum fever (OR 1.37, 95% CI 1.1–1.7), instrumental delivery (OR 1.26, 95% CI 1.09–1.46), cesarean delivery for non-reassuring fetal heart rate (OR 2.0, 95% CI 1.68–2.46), and respiratory distress requiring mechanical ventilation (OR 2.06, 95% CI 1.19–3.56). Thin meconium was associated with adverse obstetrical, delivery, and neonatal outcomes that should receive extra neonatal care and alert the pediatrician.

## 1. Introduction

Passage of fetal meconium usually begins in the first trimester and with the innervation of the anal sphincter; around 20 weeks of gestation, it becomes infrequent [1,2,3]. Meconium contains debris, intestinal secretions, bile acids, mucus, pancreatic secretions, vernix, and desquamated cells from the fetal skin and intestine [2,4,5,6]. The incidence of meconium-stained amniotic fluid (MSAF) increases with gestational age, and reaches about 15% around the time of delivery [7,8,9,10,11,12,13,14,15,16].

Intrauterine meconium passage could represent the physiological process of the fetal gastrointestinal tract maturation. However, from 34 weeks of gestation, relaxation of the anal sphincter with in utero meconium passage has been linked to fetal distress and infection [17,18,19,20,21,22,23,24,25]. MSAF is associated with adverse neonatal outcomes, including meconium aspiration syndrome (MAS), neonatal sepsis, seizures, and neonatal intensive care unit (NICU) hospitalization [1,2]. The increased rate of adverse neonatal outcomes could be explained by inflammatory processes in the lungs, chorionic plate, and umbilical vessels that were exposed to MSAF, as well as increasing risk for microbial invasion to the amniotic cavity [26,27]. Another mechanism is related to the vasoconstrictive effect of the meconium on the umbilical vessels, that could result in non-reassuring fetal heart rate (NRFHR) [28,29]. 

Meconium thickness is usually assessed by subjective impression and categorized as thin, intermediate, or thick meconium. It has been suggested that meconium thickness correlates independently with adverse neonatal outcomes. Rodríguez Fernández et al. [30] noted: “MSAF were classified into three groups: yellow (meconium that lightly stains the amniotic fluid), green (dark green moderate staining of the amniotic fluid) and thick (opaque and thick meconium, also called “pea soup meconium”)”. Gluck et al. [31] also divided meconium staining into three categories: “Light meconium, Intermediate meconium, and Heavy meconium”. Yet, data regarding the effect of thin meconium during labor are scarce and inconclusive, and most of the studies included a relatively small sample group with thin meconium. Therefore, this study assessed maternal and delivery risk factors that are associated with thin meconium and its effects on neonatal outcomes during a trial of labor, in a large cohort of patients.

## 2. Methods

This retrospective study included all women with a singleton pregnancy who underwent a trial of labor > 24 weeks of gestational age, in a single tertiary center, from January 2014 to October 2020. 

Excluded from the study were women with thick or intermediate meconium during labor, multiple pregnancies, elective cesarean deliveries (CD), and cases of intrauterine fetal demise. Yellow, green, and dark meconium staining during labor were defined as thin, intermediate, and thick meconium, respectively. 

Based on departmental protocol, the presence of MSAF and its thickness is subjectively estimated by the midwife or the obstetrician, as thin (light), moderate, or thick (heavy) meconium, and it is noted in the electronic medical records. Women with thin meconium during labor (thin meconium group) were compared to women with clear amniotic fluid during labor (control group). Our departmental policy, in cases with spontaneous rupture of membranes with MSAF, is to initiate induction of labor with oxytocin. 

### 2.1. Data Collection

The data were retrieved from the electronic medical records of the parturient and the neonate [32]. The data collected included maternal age, gestational age at delivery, gravidity, parity, smoking, BMI (kg/m^2^), hypertensive disorders (chronic hypertension, gestational hypertension, and preeclampsia), pre-gestational diabetes mellitus (DM), and gestational DM. Birth and delivery outcomes included onset of delivery (spontaneous vs. induced labor), use of epidural anesthesia, duration of the second stage of labor, intrapartum fever (defined as 38 °C during labor), trial of labor after cesarean delivery (CD) (TOLAC) and mode of labor as CD due to non-reassuring fetal heart rate (NRFHR), or instrumental delivery. Preterm delivery was defined as spontaneous labor at <37 weeks of gestation. 

Neonatal outcomes included neonatal birth weight: small for gestational age ([SGA] defined as birthweight <10th percentile according to local growth charts [33]). Additional data collected included 1- and 5-min Apgar scores, cord pH, neonatal intensive care unit (NICU) admissions, neonatal hypoglycemia, respiratory distress with mechanical ventilation, and meconium aspiration syndrome (MAS). The clinical diagnosis of respiratory distress syndrome (RDS) with mechanical ventilation was made for preterm infants with typical respiratory difficulties. MAS was diagnosed when respiratory distress occurred soon after birth, with radiological findings that could not be explained otherwise, in the presence of meconium during labor.

### 2.2. Statistical Analysis 

All statistical analyses were performed with SPSS-28 software (IBM Corp., Armonk, NY, USA). Continuous variables were presented as means and standard deviations. Nominal data were expressed as numbers and percentages. Metric variables were analyzed with t-test and Chi-squared test was used to analyze discrete variables. A *p* value < 0.05 was considered statistically significant. A multivariable logistic regression model was applied for thin meconium and adjusted for maternal age, parity, week of gestational age, epidural anesthesia, and neonatal birth weight.

### 2.3. Ethics Approval 

The study was approved by the Meir Medical Center Ethics Committee in September 2021, approval number 0167-21-MMC. Due to the retrospective nature of the study, informed consent was not required.

## 3. Results

During the study period, 31,536 women with a singleton pregnancy underwent a trial of labor and met the inclusion criteria. Among them 1946 (6.2%) women had thin meconium during labor, and 29,590 (93.8%) had clear amniotic fluid (Figure 1).

### 3.1. Demographic Characteristics 

Table 1 presents the maternal characteristics of the study groups.

There were no differences between the groups in the rates of nulliparity, DM, hypertensive disorders, obesity (BMI kg/m^2^ ≥ 30), or maternal smoking. Compared to the controls, the thin meconium group was characterized by older maternal age (31.2 ± 5.2 years vs. 30.6 ± 5.3 years, *p* < 0.001), lower rate of preterm labor (24, 1.2% vs. 1243, 4.2%, *p* < 0.001), higher rate of deliveries > 41 weeks (376, 19.3% vs. 3797, 14.8%, *p* < 0.001), and higher rate of TOLAC (124, 6.4% vs. 1565, 5.3%, *p* = 0.04).

### 3.2. Labor and Delivery Characteristics 

Table 2 shows the labor and delivery characteristics of the study groups.

The thin meconium group was characterized by a longer duration of the second stage of labor (68 ± 72 min vs. 63 ± 69 min, *p* = 0.006), increased rates of intrapartum fever (92, 4.7% vs. 975, 3.3%, *p* = 0.001), instrumental deliveries (240, 12.3% vs. 2866, 9.7%, *p* < 0.001), and CD due to NRFHR (134, 7.4% vs. 2086, 9.5%, *p* = 0.003) as compared to the control group. There were lower rates of induction in the thin meconium group (396, 20.3% vs. 7019, 23.7%, *p* = 0.001). The rates of epidural anesthesia, nuchal cord, and true knot in the umbilical cord, were similar between the groups. 

### 3.3. Neonatal Outcomes

As compared to controls, higher neonatal birthweight (3350 ± 434 vs. 3254 ± 450, *p* < 0.001) and higher rate of birth weight >4000 g (131, 6.7% vs. 1329, 4.5%, *p* < 0.001) were observed in the thin MSAF group. MAS was diagnosed in eight neonates in the thin meconium group and in none of the controls (0.41%, *p* < 0.001). The rate of respiratory distress requiring mechanical ventilation was higher in the thin meconium group as compared to controls (13, 0.67% vs. 119, 0.4%, *p* = 0.05). There were no differences between the groups regarding the incidence of SGA, 5-min Apgar score <7, cord pH, the use of phototherapy, NICU admissions, and neonatal hypoglycemia (Table 3).

### 3.4. Multivariable Logistic Regression

Multivariate logistic regression was applied for thin meconium. It was found that intrapartum fever, instrumental delivery, CD for NRFHR, MAS, and respiratory distress requiring mechanical ventilation were all independently associated with thin meconium during labor (Table 4).

## 4. Discussion

The results of the present study demonstrate that the maternal adverse outcomes of intrapartum fever, CD due to NRFHR, and instrumental delivery were independently associated with **thin** meconium during labor. Additionally, neonatal adverse outcomes of respiratory distress that required mechanical ventilation and meconium aspiration syndrome (MAS) were also found to be independently associated with thin meconium during labor.

While the association between MSAF and adverse outcomes is well-established, the effect of **thin** meconium on maternal and neonatal outcomes is less clear. It was suggested that thin meconium is indicative of chronic hypoxemic stress and thick meconium indicates acute stress [34]. Thick meconium was also found to be associated with higher neonatal infection rates [35,36]. A few studies have reported that thin meconium was *not* found to be associated with higher rates of adverse neonatal outcomes and its presence was not related to increased rates of CD [37]. Others found that thin meconium was related to a greater incidence of instrumental deliveries, but not to CD due to NRFHR or to increased rates of intrapartum fever [31]. 

Similar to other reports [31,38], the rate of thin meconium in the current study was 6.2%. Regarding associations between obstetrical risk factors and thin meconium, although older maternal age characterized the thin meconium group, we did not observe an association between maternal diseases, such as DM or hypertensive disorders and the presence of thin meconium during labor. This could be because a proactive approach to induce labor in complicated pregnancies resulted in fewer cases of MSAF. 

On the other hand, similar to previous reports [39,40], the rate of deliveries ≥41 weeks of gestation was higher in the thin meconium group. This association could be explained by motilin secretion in the neonatal gastrointestinal tract. As gestational age increases, motilin levels rise and fetal peristalsis is enhanced [8]. Notably, the relatively lower rate of induction of labor among patients with thin meconium is compatible with the finding of higher rate of deliveries >41 weeks in this group of patients. These findings reinforce the clinical approach supporting labor induction at 39 weeks [41], thus reducing the rate of MSAF and its potential complications during labor. 

In relation to intrapartum risk factors, we observed that intrapartum fever, prolonged second stage of labor, instrumental deliveries, and CD due to NRFHR were all associated with thin meconium during labor. These intrapartum factors probably reflect abnormal labor patterns that possibly increase maternal and neonatal stress. 

In the current study, the rate of MAS among neonates with thin meconium was 0.4%. Others reported a rate of MAS with thin meconium at rates ranging from 0.5% to 8.7% [31,38]. MAS is known to be associated with long-term neonatal morbidity and mortality. Therefore, although the rate of MAS is low among cases with thin meconium, its presence deserves extra neonatal care and pediatricians should be informed. 

The strengths of the current study should be noted. First, the large study population allowed us to focus on pregnancies complicated with thin meconium and to adjust for various confounders. Second, data are from a single medical center with uniform protocols. The study limitations should be noted as well. First, due to its retrospective design certain data or variables of interest were missing. Second, because the diagnosis of thin meconium is subjective, it might vary among different care givers. Third, the study lacks long-term neonatal outcome data. 

In conclusion, several maternal and neonatal adverse outcomes are independently associated with thin meconium during labor. Thin meconium is associate with MAS. Furthermore, since delivery >41 weeks is a modifiable risk factor, further studies are needed to assess the impact of labor induction at early or full term to prevent possible maternal and neonatal complications associated with MSAF.

## Figures and Tables

**Figure 1 children-10-00215-f001:**
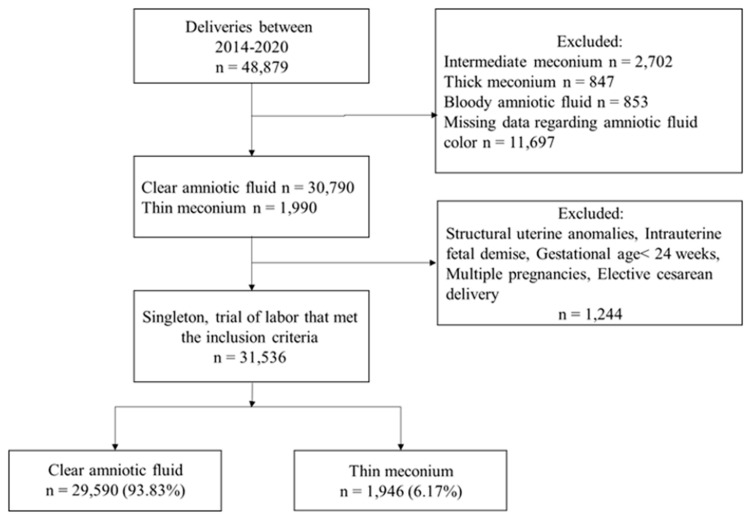
Study flow diagram.

**Table 1 children-10-00215-t001:** Baseline characteristics.

Characteristic	Thin Meconium (n = 1946)	Control (n = 29,590)	*p*-Value
Maternal age (years)	31.2 ± 5.2	30.6 ± 5.3	<0.001
Gestational age (weeks)	39.4 ± 1	39.0 ± 2	0.107
Preterm delivery (<37 weeks) (n, %)	24, 1.2	1243, 4.2	<0.001
Delivery ≥ 41 weeks (n, %)	376, 19.3	3797, 14.8	<0.001
Nulliparous (n, %)	764, 39.2	10,989, 37.1	0.059
BMI > 30 (n, %)	136, 7	1565, 6	0.162
Diabetes (n, %)	182, 9.3	3073, 10.4	0.147
Smoking (n, %)	116, 6	1530, 5.2	0.129
TOLAC (n, %)	124, 6.4	1565, 5.3	0.04
Hypertensive disorders (n, %)	50, 2.6	920, 3.1	0.181
Male fetus (n, %)	971, 49.9	14,523, 49.1	0.493

Data are presented as mean ± SD or n (%). BMI, body mass index (kg/m2); DM includes pre-gestational and gestational diabetes; TOLAC, trial of labor after cesarean section; hypertensive disorders include gestational hypertension, chronic hypertension, and preeclampsia.

**Table 2 children-10-00215-t002:** Labor and delivery characteristics.

Characteristic	Thin Meconium (n = 1946)	Control (n = 29,590)	*p*-Value
Labor induction (n, %)	396, 20.3	7019, 23.7	0.001
Epidural (n, %)	1326, 73	16,373, 74.4	0.168
Intrapartum fever (n, %)	92, 4.7	975, 3.3	0.001
Second stage duration (min)	68 ± 72	63 ± 69	0.006
Instrumental delivery (n, %)	240, 12.3	2866, 9.7	<0.001
CD due to NRFHR	135, 6.9	985, 3.3	<0.001
Nuchal cord (n, %)	523, 26.9	7629, 25.8	0.083
True knot (n, %)	29, 1.5	402, 1.3.	0.538

Data are presented as mean ± SD, or n (%), CD, cesarean delivery; NRFHR, non-reassuring fetal heart rate.

**Table 3 children-10-00215-t003:** Neonatal outcomes of the study groups.

Outcome	Thin Meconium (n = 1946)	Control (n = 29,590)	*p*-Value
Neonatal birthweight (g)	3350 ± 434	3254 ± 450	<0.001
SGA (n, %)	135, 6.9	2140, 7.2	0.627
Birth weight > 4000 g (n, %)	131, 6.7	1329, 4.5	<0.001
Apgar score at 5 min < 7 (n, %)	7, 0.36	85, 0.29	0.566
Cord blood pH ≤ 7 (n, %)	4, 0.21	75, 0.25	0.242
Phototherapy (n, %)	74, 3.8	1397, 4.7	0.174
NICU admissions (n, %)	23, 1.2	432, 1.5	0.992
Neonatal hypoglycemia (n, %)	10, 0.5	152, 0.5	0.86
RD with mechanical ventilation (n, %)	13, 0.67	119, 0.4	0.05
MAS (n, %)	8, 0.41	0	<0.001
Convulsions (n, %)	1, 0.05	11, 0.04	0.74

Data are presented as mean ± SD or n (%). SGA, small for gestational age; NICU, neonatal intensive care unit; RD, respiratory distress; MAS, meconium aspiration syndrome.

**Table 4 children-10-00215-t004:** Multivariate analysis model for adverse outcomes associated with thin meconium.

Variable	Odds Ratio	95% Confidence Interval		*p*-Value
		Lower	Upper	
Intrapartum fever	1.378	1.105	1.718	0.004
Instrumental delivery	1.263	1.092	1.461	0.002
Cesarean delivery due to non- reassuring fetal heart rate	2.037	1.685	2.463	<0.001
Respiratory distress with mechanical ventilation	2.062	1.192	3.565	0.01

## Data Availability

Data are available upon request.

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
