# Peer review of "Impact of Thin Meconium on Delivery and Early Neonatal Outcomes"

_children, 2023, doi:10.3390/children10020215_

Round 1

Reviewer 1 Report

Thank you for the opportunity to review this paper.

This is a well written paper about an interesting topic, the association of thin meconium with clinical variables and neonatal outcomes. The manuscript is clear and easy to read. I have a couple comments the authors may want to consider:

 - A main limitation of the paper is that grading of the meconium (thin, intermediate, thick) was subjective. The authors duly acknowledge this limitation in the Discussion section. Meconium is frequently graded and reported as grade I  - II - III. Does this correspond to the authors' local grading? If yes, they should say so. If not, please provide some info on local guideline about what is considered "thin", "intermediate" and "thick"

- The authors test a large number of variables in for p values. Have they corrected their p values for multiple testing?

- "MAS was diagnosed in 8 neonates of the thin meconium group and none in the controls (0.41%, p<0.001)" - this is not really surprising, as without meconium stained fluid clinicians would be very unlikely to give the diagnosis os MAS. X-rays were not reported blindly but in the knowledge of the presence of meconium.

- "This association could be explained by motilin secretion in the neonatal gastrointestinal tract. As gestational age increases, motilin levels rise and fetal peristalsis is enhanced [6]" - an additional possibility is that with increasing gestation the placenta is also aging, resulting in more fetal hypoxia and more frequent reactive passage of meconium .

- " further studies are needed to assess the impact of labor induction at early or full term to prevent possible maternal and neonatal complications associated with MSAF." - I would be reluctant to suggest induction at early term based on these findings, even to do a study. It is been well documented that elective delivery even at 37-38 weeks is associated with increased risk of neonatal complications, which may well exceed the odd ratios seen in this study.

Author Response

Reviewer 1:

Comment- English language and style are fine/minor spell check required.

Response: We thank the reviewer for his comment. The manuscript was checked and underwent extensive English revision and editing by a medical editor.

Comment- On the first row of the second page, please correct to ”... result in non-reassuring fetal heart rate ...”

Response: Thank you for the note. The sentence was corrected, as recommended:

Another mechanism is related to the vasoconstrictive effect of the meconium on the umbilical vessels, resulting in non-reassuring fetal heart rate (NRFHR) (page 2)

Comment- I am baffled by the categories the authors use to describe meconium (I didn’t even know there was the intermediate thickness of meconium – probably an oversight on my part...) – even considering the relative subjectivity of the matter, it would be so nice if the authors at least made an attempt of defining the three main categories of meconium

Response: Thank you for the comment. In accordance with our departmental protocol, the color of the amniotic fluid is divided into one of these 3 categories and is recorded as such in the medical record by the midwife. Various articles mention a similar division. Rodríguez Fernández et al. [16] noted: "MSAF were classified into three groups: yellow (meconium that lightly stains the amniotic fluid), green (dark green moderate staining of the amniotic fluid) and thick (opaque and thick meconium, also called “pea soup meconium”)". Gluck et al. [17] also divided meconium staining to three categories: "Light meconium (LM group), Intermediate meconium (IM group), and Heavy meconium (HM group)".

Comment - In the first paragraph of the Data collection section, there is a mention of a previous description of data retrieval. It is not enough for me, I am not interested in searching a previous article by the same team, on a different subject and, I might add, I consider it a lame attempt at self-citation – the authors should describe in the same amount of detail from the previous article the method they use for data retrieval.

Response: Thank you for the comment. The reference was deleted, and the paragraph was revised as recommended: "Data were retrieved from the electronic medical records of the parturient and the neonate. Missing data were completed by the investigators reviewing the medical records. Data collected included maternal age, gestational age at delivery, gravidity, parity, smoking, BMI (kg/m2), hypertensive disorders (chronic hypertension, gestational hypertension and preeclampsia), pre-gestational diabetes mellitus (DM) and gestational DM. Birth and delivery outcomes included onset of delivery (spontaneous vs. induced labor), use of epidural anesthesia, duration of the second stage of labor, intrapartum fever (defined as 38°C during labor), trial of labor after CD (TOLAC) and mode of delivery (CD due to non-reassuring fetal heart rate (NRFHR) or instrumental delivery). Preterm delivery was defined as spontaneous labor at <37 weeks of gestation. Neonatal outcomes included neonatal birth-weight, small for gestational age ([SGA] defined as birthweight <10th percentile according to local growth charts (1). Additional data included Apgar scores, cord pH, neonatal intensive care unit (NICU) admissions, neonatal hypoglycemia (blood glucose < 40 mg/dL), treatment with phototherapy, convulsions, respiratory distress (RD) with mechanical ventilation and meconium aspiration syndrome (MAS). The clinical diagnosis of RDS with mechanical ventilation was made for preterm infants with typical respiratory difficulties. MAS was diagnosed when respiratory distress occurred soon after birth, with radiological findings that could not be explained otherwise, in the presence of meconium during labor."

Comment - In Figure 1, the authors should explain the abbreviation IUFD, and state ”weeks” after "”Gestational age < 24”.

Response: We thank the reviewer for the comment. Attached the revised figure:

Comment- where the authors describe and corelate the data they collected, there is no mention of aspiration of clear amniotic fluid. In all seven years they investigated, there was no case of aspiration of clear amniotic fluid during delivery?

Response: We agree with the reviewer that clear amniotic fluid aspiration might be
part of the neonatal RD cases we describe in table 3. However, as this entity
is rare and our data collection was retrospective in nature, we followed the
neonatal diagnosis as was given during NICU hospitalization by the
neonatologists. In the 119 cases of respiratory distress with clear amniotic
fluid (ie control) no one was diagnose with clear amniotic fluid aspiration.
We add a sentence in the discussion " In the current study, the rate of MAS
and mechanical ventilation among neonates with thin meconium was 0.4% with
odds ratio of 2 (p=0.01) as compared to clear amniotic fluid study group
Others reported of MAS in thin meconium in a rate that ranges between
0.5-8.7% [16,27].  MAS is known to be associated with long term neonatal
morbidity and mortality, thus although its rate is low among cases with thin
meconium, its presence deserve extra neo-natal care and should alert the
pediatrician.

Comment- On the first row of the Neonatal outcomes section, please add ”g” to units expressing birthweight, and also use ”g” not ”gr” on the second row.

Response: The sentence was revised as recommended: "Compared to controls, higher neonatal birthweight (3350 ± 434 g vs. 3254 ± 450 g, p<0.001) and higher rate of birth weight > 4000 g"

Comment- In Table 4, the authors should explain the abbreviations OR and RD.

Response: The revised Table is attached.

Table 4. Multivariate analysis model for adverse outcomes associated with thin meconium

Variable

Odds ratio

95% Confidence interval

p-value

Lower

Upper

Intrapartum fever

1.378

1.105

1.718

0.004

Instrumental delivery

1.263

1.092

1.461

0.002

Cesarean delivery due to non- reassuring fetal heart rate

2.037

1.685

2.463

<0.001

Respiratory distress with mechanical ventilation

2.062

1.192

3.565

0.01

Comment- In the second paragraph of the Discussions section, there is a phrase (Few studies.... increasing CD rates) that needs reformulation, in order to make it clearer.

Response: Thank you for the comment, the paragraph was revised: A few studies have reported that thin meconium was not found to be associated with higher rates of adverse neonatal outcomes and its presence was not related to increased rate of CD.

Reviewer 2 Report

The manuscript is rather interesting and involves a topic which is rarely researched, despite the various complications it can produce.

First of all, it would have been better if the authors left the line numbers on the side of the page, it is easier for both reviewers and editors to point out exactly where the problem lies. I’ll do my best anyway.

On the first row of the second page, please correct to ”... result in non-reassuring fetal heart rate ...”

I am baffled by the categories the authors use to describe meconium (I didn’t even know there was the intermediate thickness of meconium – probably an oversight on my part...) – even considering the relative subjectivity of the matter, it would be so nice if the authors at least made an attempt of defining the three main categories of meconium.

In the first paragraph of the Data collection section, there is a mention of a previous description of data retrieval. It is not enough for me, I am not interested in searching a previous article by the same team, on a different subject and, I might add, I consider it a lame attempt at self-citation – the authors should describe in the same amount of detail from the previous article the method they use for data retrieval.

In Figure 1, the authors should explain the abbreviation IUFD, and state ”weeks” after ”Gestational age < 24”.

On page 5, where the authors describe and corelate the data they collected, there is no mention of aspiration of clear amniotic fluid. In all seven years they investigated, there was no case of aspiration of clear amniotic fluid during delivery?

On the first row of the Neonatal outcomes section, please add ”g” to units expressing birthweight, and also use ”g” not ”gr” on the second row.

In Table 4, the authors should explain the abbreviations OR and RD.

In the second paragraph of the Discussions section, there is a phrase (Few studies.... increasing CD rates) that needs reformulation, in order to make it clearer.

Author Response

Reviewer 2:

Comment- Extensive editing of English language and style required

Response: The manuscript was edited by a medical editor.

Comment- Please explain about logistic regression method in more detail.

Response: We thank the reviewer for the comment. The paragraph regarding the multivariable analysis in the method was revised: "Multivariable logistic regression model estimated the 95% confidence intervals and the adjusted odds ratios for potential risk factors for thin meconium. The analysis was applied for thin meconium and adjusted for maternal age, parity, week of gestational age, epidural, and neonatal birthweight. "

Comment- Explain all the abbreviations in tables.

Response: All abbreviations were explained.

Reviewer 3 Report

In their paper, Schreiber et al., determined the impact of thin meconium on delivery and early neonatal outcome. The analysis showed that intrapartum fever, instrumental delivery, caesarean delivery due to non-reassuring fetal heart rate, and respiratory distress requiring mechanical ventilation were associated with thin meconium. After thoroughly reading the manuscript, I have some minor concerns. I would propose the minor revision of this manuscript and please see my specific comments below.

Minor:

Please include all the necessary stats such as t-values etc.

Please explain about logistic regression method in more detail.

Explain all the abbreviations in tables.

Author Response

Minor:

Comment: Please include all the necessary stats such as t-values etc.

Answer: We thank the reviewer for the comment. We included all the necessary stats. The paragraph regarding the multivariable analysis in the method was revised: "Multivariable logistic regression model estimated the 95% confidence intervals and the adjusted odds ratios for potential risk factors for thin meconium. The analysis was applied for thin meconium and adjusted for maternal age, parity, week of gestational age, epidural, and neonatal birthweight. "

Please explain about logistic regression method in more detail.

Answer: Table 4. Multivariate analysis model for adverse outcomes associated with thin meconium

Variable

Odds ratio

95% Confidence interval

p-value

Lower

Upper

Intrapartum fever

1.378

1.105

1.718

0.004

Instrumental delivery

1.263

1.092

1.461

0.002

Cesarean delivery due to non- reassuring fetal heart rate

2.037

1.685

2.463

<0.001

Respiratory distress with mechanical ventilation

2.062

1.192

3.565

0.01

Explain all the abbreviations in tables.

Response: All abbreviations were explained.